# The Role of Statins on *Helicobacter pylori* Eradication: Results from the European Registry on the Management of *H. pylori* (Hp-EuReg)

**DOI:** 10.3390/antibiotics10080965

**Published:** 2021-08-11

**Authors:** María Caldas, Ángeles Pérez-Aisa, Bojan Tepes, Alma Keco-Huerga, Luis Bujanda, Alfredo J. Lucendo, Luis Rodrigo, Dino Vaira, Luis Fernández-Salazar, Jose M. Huguet, Jorge Pérez-Lasala, Natasa Brglez Jurecic, Galina Fadeenko, Jesús Barrio, Miguel Areia, Juan Ortuño, Rinaldo Pellicano, Marcis Leja, Javier Molina-Infante, Pavel Bogomolov, Sergey Alekseenko, Manuel Domínguez-Cajal, Judith Gómez-Camarero, Vassiliki Ntouli, Samuel J. Martínez-Domínguez, Rafael Ruiz-Zorrilla, Oscar Núñez, Aiman Silkanovna Sarsenbaeva, Pedro Almela, Perminder Phull, Marta Espada, Ignasi Puig, Olga P. Nyssen, Francis Mégraud, Colm O’Morain, Javier P. Gisbert, on behalf of the Hp-EuReg Investigators

**Affiliations:** 1Centro de Investigación Biomédica en Red de Enfermedades Hepáticas y Digestivas (CIBEREHD), Gastroenterology Unit, Instituto de Investigación Sanitaria Princesa (IIS-IP), Hospital Universitario de La Princesa, Universidad Autónoma de Madrid (UAM), 28006 Madrid, Spain; maria.caldas@salud.madrid.org (M.C.); martaespadasanchez@gmail.com (M.E.); opn.aegredcap@aegastro.es (O.P.N.); 2Digestive Unit, Hospital Costa del Sol and Red de Investigación en Servicios de Salud en Enfermedades Crónicas (REDISSEC), 29651 Marbella, Spain; drapereza@hotmail.com; 3AM DC Rogaska, 3250 Rogaska Slatina, Slovenia; bojan.tepes@siol.net; 4Department of Gastroenterology, Hospital de Valme, 41014 Sevilla, Spain; almakh94@hotmail.com; 5Centro de Investigación Biomédica en Red de Enfermedades Hepáticas y Digestivas (CIBERehd), Department of Gastroenterology, Hospital Donostia/Instituto Biodonostia, Universidad del País Vasco (UPV/EHU), 20014 San Sebastián, Spain; luis.bujanda@osakidetza.net; 6Centro de Investigación Biomédica en Red de Enfermedades Hepáticas y Digestivas (CIBEREHD), Instituto de Investigación Sanitaria Princesa (IIS-IP), Department of Gastroenterology, Hospital General de Tomelloso, 13700 Ciudad Real, Spain; ajlucendo@sescam.jccm.es; 7Gastroenterology Unit, Hospital Universitario Central de Asturias, 33011 Oviedo, Spain; lrrodrigo@uniovi.es; 8S. Orsola Malpighi Hospital, 40138 Bologna, Italy; berardino.vaira@unibo.it; 9Digestive Service, Hospital Clínico Universitario de Valladolid, 47003 Valladolid, Spain; lfernandezsa@saludcastillayleon.es; 10Gastroenterology Unit, Consorcio Hospital General Universitario de Valencia, 46014 Valencia, Spain; huguet_jos@gva.es; 11Gastroenterology Unit, University Hospital of Torrejón, 28850 Madrid, Spain; jperez@torrejonsalud.com; 12Diagnosticni Center Bled d.o.o., 4260 Bled, Slovenia; natasa.brglez.jurecic@gmail.com; 13Digestive Ukrainian Academy of Medical Sciences, 01030 Kyiv, Ukraine; g.fadeenko@gmail.com; 14Gastroenterology Unit, Hospital Universitario Río Hortega, Gerencia Regional de Salud de Castilla y León (SACYL), 47012 Valladolid, Spain; jbarrio@saludcastillayleon.es; 15Portuguese Oncology Institute, 3000-075 Coimbra, Portugal; miguel.areia@ipocoimbra.min-saude.pt; 16Digestive Service, Hospital Universitari i Politècnic La Fe, 46026 Valencia, Spain; jortunoc@comv.es; 17Molinette Hospital, Città della Salute e della Scienza di Torino, 10126 Turin, Italy; rpellicano@cittadellasalute.to.it; 18Digestive Diseases Center, GASTRO, Institute of Clinical and Preventive Medicine and Faculty of Medicine, University of Latvia, LV-1079 Riga, Latvia; cei@latnet.lv; 19Department of Gastroenterology, Hospital San Pedro de Alcántara, 10003 Cáceres and Centro de Investigación Biomédica en Red de Enfermedades Hepáticas y Digestivas (CIBERehd), 28006 Madrid, Spain; javier.molinai@salud-juntaex.es; 20Universal Clinic Private Medical Center, 129110 Moscow, Russia; bpo73@list.ru; 21Far Eastern State Medical University, 680000 Khabarovsk, Russia; nauka@mail.fesmu.ru; 22Digestive Service, Hospital San Jorge, 22004 Huesca, Spain; mdominguezc@salud.aragon.es; 23Department of Gastroenterology and Hepatology, Complejo Asistencial Universitario de Burgos, 09006 Burgos, Spain; jgomezcam@saludcastillayleon.es; 24General Hospital Pireaus, 185 36 Pireaus, Greece; vassiliki.ntouli@gmail.com; 25Digestive Service, Hospital Clínico Universitario Lozano Blesa and CIBERehd, 50009 Zaragoza, Spain; martinezdominguezsamuel@gmail.com; 26Digestive Service, Hospital de Sierrallana Torrelavega, 39300 Cantabria, Spain; rafael.ruiz@scsalud.es; 27Digestive Service, Hospital Universitario Sanitas La Moraleja, 28050 Madrid, Spain; onumar@gmail.com; 28Gastroenterologist Department of Regional Clinical Hospital №3, 454076 Chelyabinsk, Russia; aiman-ss@yandex.ru; 29Digestive Service, Hospital Universitari General de Castelló, 12004 Castellón, Spain; almela_ped@gva.es; 30Aberdeen Royal Infirmary, Aberdeen AB25 2ZN, UK; p.s.phull@abdn.ac.uk; 31Digestive Service, Althaia Xarxa Assistencial Universitària de Manresa and Universitat de Vic-Universitat Central de Catalunya (UVicUCC), 08242 Manresa, Spain; ipuig@althaia.cat; 32Laboratoire de Bactériologie, Hôpital Pellegrin, Bordeaux & INSERM U1053 BaRITOn, Université de Bordeaux, 33076 Bordeaux, France; francis.megraud@chu-bordeaux.fr; 33Department of Clinical Medicine, Trinity College Dublin, D24 NR0A Dublin, Ireland; colmOMorain@rcpi.ie

**Keywords:** statins, *Helicobacter pylori*, treatment

## Abstract

Statins could increase the effectiveness of *Helicobacter pylori* eradication therapies due to their anti-inflammatory effect. The aim of this study was to analyze the impact of this therapeutic association in real life. This is a multicenter, prospective, non-interventional study aimed at evaluating the management of *H. pylori* by European gastroenterologists. Patients were registered in an e-CRF by AEG-REDCap from 2013 to 2020. The association between statin use and *H. pylori* eradication effectiveness was evaluated through multivariate analysis. Overall, 9988 and 705 patients received empirical and culture-guided treatment, respectively. Overall, statin use was associated with higher effectiveness in the empirical group (OR = 1.3; 95%CI = 1.1–1.5), but no association was found with first-line treatment effectiveness (N = 7738); as an exception, statin use was specifically associated with lower effectiveness of standard triple therapy (OR = 0.76; 95%CI = 0.59–0.99). In the rescue therapy empirical group (N = 2228), statins were associated with higher overall effectiveness (OR = 1.9; 95%CI = 1.4–2.6). However, sub-analyses by treatment schemes only confirmed this association for the single-capsule bismuth quadruple therapy (OR = 2.8; 95%CI = 1.3–5.7). No consistent association was found between statin use and *H. pylori* therapy effectiveness. Therefore, the addition of statins to the usual *H. pylori* treatment cannot be currently recommended to improve cure rates.

## 1. Introduction

*Helicobacter pylori* (*H. pylori*) is a Gram-negative bacterium involved in the etiopathogenesis of several common gastric diseases such as peptic ulcer, chronic gastritis, or gastric cancer, but also several extra-gastric diseases (e.g., iron deficiency anemia, idiopathic thrombocytopenic purpura, and vitamin B12 deficiency), and the list of possible associations is constantly increasing [1,2,3].

Despite numerous attempts, an effective, unique, and global eradication therapy (providing ≥90% success) has not been found, mainly due to the geographical variability of antibiotic resistance rates and their increase worldwide [1,4,5].

Several strategies have been proposed to optimize the success rate of eradication therapies, most of them focused on extending treatment duration, using more potent drugs to decrease gastric acidity (such as high doses of proton pump inhibitors (PPIs)), or using quadruple instead of triple therapies. Other strategies, such as the use of dual therapy combining PPI and amoxicillin administered four times daily for 14 days, have provided good results in the Asiatic setting but need to be evaluated specifically in the European context [5,6,7,8,9,10,11].

One strategy suggested to increase eradication effectiveness has been the addition of statins. Statins mainly inhibit the activity of 3-hydroxy-3-methylglutaryl coenzyme A (HGM-CoA) reductase, blocking the first step of the L-mevalonate pathway and reducing the production of cholesterol in hepatocytes and other tissues. This causes a drop in LDL-cholesterol levels, explaining the main role of these drugs in the prevention of cardiovascular diseases [12,13]. However, by inhibiting this step, other metabolites derived from the cholesterol synthesis cascade such as non-steroidal isoprenoid compounds (named farnesyl pyrophosphate (FPP) and geranylgeranyl pyrophosphate (GGPP)) are affected. These molecules play a role in the post-translational lipid modification of several proteins involved in guanosine triphosphatase (GTPase) activation and intracellular signaling. This is known as the “pleiotropic effect” of statins, which does not seem to be proportional to the magnitude of the decline in LDL-cholesterol levels [14,15,16].

This pleiotropic effect has been associated with anti-inflammatory properties involving infectious and autoimmune diseases or even neoplastic conditions, although definitive conclusions have not been obtained [17,18,19,20,21,22,23]. In addition, a hypothetical role in the healing of gastric inflammation has also been suggested [24,25], and some authors have also proposed a possible effect increasing the effectiveness of the traditional eradication therapies against *H. pylori* infection [26]. Little is known about the exact mechanism of action of these drugs on *H. pylori* infection. Some authors have suggested that the reduction in cellular cholesterol caused by the use of statins could result in reduced VacA activity and attenuated CagA-induced inflammation in gastric cells and in the induction of autophagy in macrophages infected with *H. pylori* [27,28]. However, in a previously published study performed in mice, *H. pylori* viability was not reduced by pravastatin, even though a drop in inflammation in gastritis could be observed [24].

Considering the limited evidence in this field and the current necessity to optimize eradication therapies, we evaluated a cohort of patients from the long-term prospective clinical practice European Registry on the Management of *H. pylori* (Hp-EuReg) [29], with the aim of obtaining updated information on the role of statins on *H. pylori* eradication.

## 2. Results

From May 2013 to January 2020, 10,915 cases participating in the Hp-EuReg were selected for analysis, based on whether they were or were not daily statin receivers as part of a chronic therapy prescribed for cardiovascular risk prevention. Quality criteria were applied, leading to a final number of 10,693 patients included for analysis (Figure 1). These patients came from 123 European hospitals from a total of 25 different countries (online Appendix A).

Patients were divided in two groups according to statin use: 2635 cases received statins during the *H. pylori* eradication treatment and 8058 patients were not statin-consumers. Both cohorts were heterogeneous concerning the following variables: gender (female vs. male), presence of allergy to penicillin (yes vs. no), indication for *H. pylori* diagnosis (no ulcer vs. ulcer disease), use of different PPI doses (low vs. standard vs. high), and length of eradication treatment (7 vs. 10 vs. 14 days) (Table 1). Other reported basal characteristics were similar in both cohorts: line of eradication treatment prescribed (first vs. rescue attempts), type of prescription administered (empirical vs. culture-guided), treatment compliance (<90% vs. ≥90%), and presence vs. absence of adverse events (AEs) (Table 1).

The type of statin was reported in 13% of all statins consumers and were mainly: simvastatin (*n* = 155, 45%), atorvastatin (*n* = 134, 39%), rosuvastatin (*n* = 37, 11%), and a last group named “others” (*n* = 17, 5%) including other statins (e.g., pravastatin, pitavastatin, lovastatin, or fluvastatin) (Table 2). Statin dosages were not reported and therefore no further analysis could be performed based on this variable. The distribution of the eradication therapies administered according to the statin used is shown in online Appendix A.

### 2.1. Empirical Prescription

Overall, 93% of the patients received an empirical prescription to treat *H. pylori* (*n* = 9988). Higher eradication rates were reported in the statin cohort than in non-statin users (88% vs. 85%; *p* < 0.001 in the univariate analysis, and an OR = 1.27; 95% CI: 1.1–1.5; *p* < 0.05 in the multivariate analysis). Other variables included in the multivariate analysis were also shown to be associated with an increase in mITT effectiveness, including good compliance, prescription of a first eradication attempt instead of a rescue regimen, male gender, presence of gastric ulcer disease, 10 and 14-day treatments (instead of 7-day treatment), and use of standard or high PPI doses (instead of low doses) (Table 3 and Table 4).

No differences were found in the incidence rates of AEs: 24% vs. 23% in statin users and non-users, respectively.

Sub-analysis by treatment scheme in first and rescue therapy is described below.

#### 2.1.1. First-Line Therapy

A total of 7738 patients received an empirical first-line therapy, of which 1875 patients were daily statin users and 5863 were non-users. No difference was found in terms of overall effectiveness, according to statin use, which was 88.5% in statin users and 87% in non-users (Table 5). The multivariate analysis showed no association between statin status and mITT effectiveness (OR = 1.1; 95% CI: 0.9–1.3; *p* = 0.286). The variables that showed significant results in this context are shown in Table 6.

The standard triple regimen containing a PPI, clarithromycin, and amoxicillin, and the non-bismuth quadruple concomitant therapy (PPI-clarithromycin-amoxicillin-nitroimidazole), were the therapies most frequently prescribed in both groups.

The sub-analysis by therapy revealed that the highest eradication rates were obtained with the single capsule bismuth quadruple therapy (containing tetracycline, metronidazole, and bismuth salts in a single capsule administered together with a PPI), the bismuth-amoxicillin-clarithromycin quadruple therapy (adding these antibiotics to a PPI), and the non-bismuth concomitant regimen (PPI-amoxicillin-clarithromycin-nitroimidazole); all of them exhibited close to 90% mITT effectiveness (Figure 2). Similar results were obtained for these therapies in statin users and non-users, with non-significant differences between both cohorts. However, the statin users who received the concomitant quadruple regimen showed a tendency towards higher effectiveness than non-users that was close to statistical significance (OR = 1.4; 95% CI: 1–2; *p* = 0.059). Only the standard triple therapy showed a significant difference in effectiveness according to the statin status, with lower effectiveness in statin users (OR = 0.763; 95% CI: 0.59–0.99; *p* < 0.05) (Table 5 and Table 6).

Regarding safety, the overall incidence of AEs was similar between both groups (20% in statin users vs. 21% in non-users), and most of them were mild (Figure 3). However, the rate of AEs was significantly different between users and non-users in the concomitant quadruple therapy group specifically (19% vs. 25%, *p* < 0.05) (Table 5, Figure 3, online Appendix A). The incidence rate of serious AEs (SAEs) was <1% in both groups. In the statin users group, one patient experienced diarrhea with disability after using the concomitant quadruple therapy. In the non-users cohort, three patients presented an SAE, all after the single capsule bismuth therapy. These events were: an episode of acute pancreatitis, a process of abdominal pain and vomiting, both requiring hospitalization, and a third event that was not explained in the database.

The specific analysis evaluating effectiveness or safety of eradication therapies according to the different statins used (simvastatin, atorvastatin, rosuvastatin, and other statins) showed no differences, neither in the overall group nor when considering each therapy separately (online Appendix A).

#### 2.1.2. Rescue Therapies

A total of 2228 patients received an empirical rescue therapy after failing at least one treatment attempt to eradicate *H. pylori* infection, 1612 (72%) received a second-line therapy, 459 (21%) a third-line, 111 (5%) a fourth-line, 34 (1.5%) a fifth-line, and 12 (0.5%) a sixth-line.

Likewise, two cohorts could be distinguished: statin users (*n* = 589) and non-users (*n* = 1639). Overall mITT effectiveness was found to be different in these two cohorts: 87% in the statin-users and 78% in non-users (*p* < 0.001) (Table 7). The multivariate analysis showed statin-use as a variable being significantly associated with higher mITT effectiveness (OR = 1.9; 95% CI: 1.4–2.6; *p* < 0.001). See Table 8 for detailed results of the multivariate analysis.

The triple therapy containing a PPI, amoxicillin, and levofloxacin, followed by the single capsule bismuth quadruple therapy, and the bismuth-amoxicillin-levofloxacin quadruple therapy (adding these antibiotics to a PPI) were the three most frequently prescribed therapies in both cohorts (Table 7).

The sub-analysis performed specifically on each therapy revealed that the best mITT effectiveness result was achieved both with the single capsule bismuth and the bismuth-amoxicillin-levofloxacin quadruple therapies (close to 90%; Figure 2). The presence of statins was significantly associated with higher effectiveness only in the single capsule bismuth treatment (OR = 2.8; 95% CI: 1.3–5.7; *p* < 0.05). No association between effectiveness and statin use was found within the remaining therapies (Table 8).

The overall incidence of AEs was higher in statin users than in non-users (35% vs. 28%, *p* < 0.05), and most of them were moderate (Figure 3). Only the bismuth-amoxicillin-clarithromycin quadruple therapy showed a significantly higher rate of AE in the statin cohort than in non-users; however, since the sample size of both cohorts was very small, this finding remains non-conclusive. See Table 7, Figure 3 and online Appendix A for a more specific analysis.

SAEs were present only in four patients (<1%), all of them in the non-statin users cohort: one patient presented severe abdominal pain requiring hospitalization (after using the standard triple therapy), one patient presented an episode of severe diarrhea (after the triple amoxicillin-levofloxacin therapy), one patient a Clostridioides difficile infection (after the single capsule bismuth quadruple), and the last patient presented nausea, diarrhea, weight loss, and laboratory abnormalities (after taking a marginal regimen).

The specific analysis according to the different statins used (simvastatin, atorvastatin, rosuvastatin, and other statins) showed, again, no differences both in the overall group and considering each therapy separately. A detailed analysis is shown in online Appendix A.

### 2.2. Culture-Guided Prescriptions

In total, 705 patients received an eradication therapy guided by the antibiotic resistance profile. Unlike what happened with the empirical group, the effectiveness was similar between statin users and non-users (90% vs. 87%; *p* = 0.27 in the univariate analysis; OR = 1.3; 95% CI: 0.71–2.4; *p* = 0.378 in the multivariate analysis). The only variables included in the multivariate analysis that showed significant results in terms of mITT effectiveness were the administration of a first-line instead of a rescue treatment approach, and good compliance (Table 4). No differences were observed in the rate of AEs between both groups (22% vs. 20% for statin users and non-users, respectively, *p* = 0.589) (Table 3). Detailed analyses by treatment scheme in first-line and rescue eradication therapy are described below.

#### 2.2.1. First-Line Therapy

A total of 531 patients received a first-line therapy: 130 were statin users and 401 were non-users. No differences concerning the effectiveness were found between both cohorts (93.5% for statins vs. 88% for non-users; *p* = 0.095). The multivariate analysis showed no significant differences in mITT effectiveness results according to the statin-status (OR = 1.9; 95% CI: 0.85–4.1; *p* = 0.120) (Table 9 and Table 10).

The non-bismuth quadruple sequential therapy (PPI-clarithromycin-amoxicillin-nitroimidazole administered sequentially), and the standard triple therapy were the most frequently prescribed therapies in both groups. Both therapies showed a success higher than 90%. No differences in effectiveness or safety were found between the presence or absence of statins for each specific therapy (Table 9 and Table 10, and Figure 2).

According to the overall safety, both groups experienced similar rates of AEs (21% vs. 22%), which were mostly of mild intensity (Figure 3). No SAEs were reported.

The specific analysis according to the different statins used (simvastatin, atorvastatin, rosuvastatin, and other statins) showed no association between any of them and the effectiveness or safety of the eradication therapies, either globally or in the sub-analyses of the main therapies used (online Appendix A).

#### 2.2.2. Rescue Therapies

A total of 173 patients received a rescue therapy guided by culture, 74 (43%) received a second attempt, 65 (38%) a third attempt, 19 (11%) a fourth attempt, 11 (6%) a fifth attempt, and 4 (2%) a sixth-line eradication attempt. Of these, 33 were statin users and 140 were non-users.

mITT effectiveness was similar in statin users and non-users in the univariate analysis (74% vs. 82%; *p* = 0.379). No association was found between effectiveness and any of the other variables included in the multivariate analysis. Safety findings were also similar (Figure 3).

The majority of patients in both groups received a triple levofloxacin-amoxicillin therapy or a triple therapy adding rifabutine and amoxicillin to a PPI. No differences in mITT effectiveness were found to be significant between statin users and non-users (Table 9 and Table 10, and Figure 2), or between the use of simvastatin vs. atorvastatin (online Appendix A).

## 3. Discussion

The present study is a sub-analysis of the Hp-EuReg focused on evaluating whether the use of statins could modify the effectiveness or the safety of the therapies prescribed against *H. pylori*. It is important to clarify that statins had already been initiated in our study when the eradication regimen was prescribed.

We separately evaluated empirical and culture-guided prescriptions of eradication therapies, differentiating within each group between the first treatment approach and the rescue attempts, as both conditions are considered to modify per se the effectiveness of *H. pylori* treatment.

Patients receiving an empirically prescribed first-line therapy showed optimal effectiveness rates independently of their statin-status (close to 90% in both cases). Other parameters such as the use of longer eradication therapies, use of standard or high PPI doses, or good compliance were associated with higher effectiveness. These associations had already been observed in previous reports, which reinforces our findings [30,31,32].

The specific analysis of first-line therapies showed that the single capsule bismuth quadruple, the bismuth-amoxicillin-clarithromycin quadruple, and the non-bismuth concomitant regimen obtained the highest effectiveness, in agreement with previous analyses [7,33,34]. Neither of these therapies showed an association between treatment effectiveness and the use of statins. Only in the standard triple therapy was the use of these drugs associated with lower effectiveness. This finding is opposed to what was described in the randomized studies of Nseir and Hassan, in which the addition of simvastatin 20 mg twice daily to the standard triple therapy containing clarithromycin and amoxicillin showed a statistically significant increase in effectiveness of 15–20% [26,35]. Despite their findings, several voices questioned whether this hypothetical increase would be clinically significant considering the unacceptable effectiveness rate reported for the standard triple therapy over the years in most geographical areas [1,6,36].

In the group of patients receiving an empirical rescue treatment, the use of statins was associated with higher effectiveness in the overall cohort. Other factors that showed an association with higher effectiveness were peptic ulcer diagnosis, younger age of the patients, use of longer therapies, use of high doses of PPIs, and good compliance. However, the specific sub-analysis of the main eradication therapies only revealed an association between the use of statins and the effectiveness of the single capsule bismuth quadruple therapy. No previous evidence concerning this association had been published before. However, evidence evaluating another quadruple therapy, the bismuth-amoxicillin-clarithromycin quadruple therapy, had previously provided conflicting results: Parsi and collaborators evaluated the effectiveness of adding simvastatin (10 or 20 mg once daily) to a 14-day treatment with this quadruple therapy and found no significant differences in treatment effectiveness between the statin and placebo groups, with the effectiveness being around 90% in both cases [37]. On the other hand, Sarkeshikian and collaborators found significantly better results with the addition of atorvastatin 40 mg once daily to 14-day bismuth-amoxicillin-clarithromycin quadruple therapy (increasing effectiveness from 65% to 78%) [38]. Although this increase in effectiveness might seem relevant, the effectiveness values obtained in both cases can be considered suboptimal; accordingly, possible factors explaining these insufficient results should be carefully addressed.

Therefore, the results obtained in our study concerning statins’ role in eradication therapy were not homogeneous in first or rescue lines, and neither were they when evaluating the different eradication treatment schemes. Up to now, the exact role of this hypothetical synergism between drugs based on a pure anti-infection role has not been well elucidated or universally proven. Some authors have shown that cholesterol cellular depletion caused by statins promotes autophagy in macrophages containing *H. pylori,* producing a drop in the burden of these bacteria. However, other groups failed to show this descent in bacterial viability, so more studies are needed to uncover the real effect of this association in eradication terms of the infection [24,27]. The role of the statins in decreasing the inflammatory response caused by Cag A protein from *H. pylori* in gastric mucosa, by lowering cellular cholesterol levels and decreasing NF- κβ and IL-8 has also been reported, with encouraging results in gastric cancer. Again, large scale studies need to be performed to evaluate this fact before global conclusions can be drawn [22,28].

Concerning safety parameters, the addition of statins to the eradication therapy was not associated with significant changes in AE rates (which appeared in around 20–25% of the patients). Only in the rescue-attempt group was a higher rate of AEs observed in statin-users compared to non-users (35%, vs. 28%), although this difference was not confirmed when addressing each therapy separately. In general terms, most of the AEs were mild, and the rate of SAEs was marginal in both cohorts.

Finally, no specific differences in effectiveness or safety were found according to the different statins used: simvastatin vs. atorvastatin vs. rosuvastatin vs. other minority statins. Other studies divided statins in two groups categorized as high or low potency, considering the type of drug and its dosage, as well as the magnitude of the decrease induced in basal LDL-cholesterol (≥45% or <45%, respectively) [39].

Our study has several limitations. The main one is its design as an observational non-interventional study. As such, statin use was not randomized, with consequent heterogeneous cohorts for certain demographical variables and the resulting potential risk of bias. In spite of this limitation, we believe this design provides interesting results coming from a real-practice European setting, which are, therefore, more easily applicable to daily clinical management in this area. Another limitation is the lack of information concerning the exact doses and types of statins used in more than 80% of patients, which precluded establishing subgroups according to statin potency. This is explained by the initial design of the Hp-EuReg project, which was focused on evaluating *H. pylori* treatment in routine clinical practice (far from the controlled conditions of experimental settings) rather than studying the relationship between statin types (or their doses) and the eradication regimens, especially considering the lack of previous formal evidence of this association.

In general terms, Hp-EuReg provides a large sample size suitable for evaluating strategies to increase the global effectiveness of the eradication therapies to treat *H. pylori*. Concerning statin use, although combined administration of statins with some therapies was able to modify their effectiveness or safety, this effect was not observed in other lines and/or using other eradication regimens. This lack of a common or consistent behavior raises some concerns about the real utility of adding statins, especially considering that no pathophysiological explanation for the synergistic effect with antibiotics has been unveiled in the framework of *H. pylori* treatment [27,28]. It should also be taken into consideration that the addition of another medication to an already complicated-to-follow regimen can affect treatment adherence, which is considered a relevant factor affecting effectiveness. Therefore, the addition of statins to *H. pylori* treatment should not be generally recommended to improve eradication.

## 4. Materials and Methods

The “European Registry on *H. pylori* Management” (Hp-EuReg) is an international (30 countries), multicenter (>300 investigators), prospective, non-interventional registry that was started in 2013 and was promoted by the European Helicobacter and Microbiota Study Group (www.helicobacter.org, accessed on 6 August 2021).

The Hp-EuReg protocol was approved by the Ethics Committee of La Princesa University Hospital (Madrid, Spain) [29] and was prospectively registered at ClinicalTrials.gov under the code NCT02328131. The study protocol conforms to the ethical guidelines of the 1975 Declaration of Helsinki as reflected in the approval of the institution’s human research committee. Written, informed consent was obtained from each patient included in the study.

Criteria for country selection, national coordinators, and gastroenterologist recruiting investigators are shown in the protocol publication. Monitoring (at least 10% of the included records in each country and each hospital, respectively), quality of the data, and a list of variables and outcomes are also shown in the same protocol [29].

Data were recorded in an Electronic Case Report Form (e-CRF) and collected and managed using REDCap, a research electronic data capture platform hosted at “Asociación Española de Gastroenterología” (AEG; www.aegastro.es, accessed on 6 August 2021), a non-profit Scientific and Medical Society focused on Gastroenterology research [40,41].

The aim of the current analysis was to evaluate if the concomitant use of statins prescribed for cardiovascular prevention and chronically used could modify the effectiveness rates of *H. pylori* eradication therapies. Secondary aims were directed to assess if this role would be different depending on the type of eradication therapy or statins prescribed and whether these statins would modify AE rates.

For this specific analysis, some quality criteria were applied: only patients with available information on whether they were daily statin users or not were included for evaluation. Exclusion criteria were: patients under the age of 18 years, patients with a period of less than four weeks between the end of the eradication therapy, and patients who were checked for treatment response with the use of serology.

### 4.1. Variables

The e-CRF registered 290 variables containing information about a patient’s demographics, comorbidity, indication and method of diagnosis, previous eradication regimens prescribed (if any), current treatment, and data on effectiveness, safety, and compliance. The variable treatment length was assessed using three categories, corresponding to the most frequent treatment durations: 7, 10, and 14 days. Similarly, PPI data were standardized using PPI acid inhibition potency as defined by Kirchheiner [42] and Graham [32], and classified as low, standard and high dose PPIs (online Appendix A). Eradication was confirmed with at least one of the following diagnostic methods: urea breath test, stool antigen test and/or histology.

### 4.2. Effectiveness Analysis

Treatment eradication rate was the main outcome, and was studied in two sets of patients as follows: a modified intention-to-treat (mITT) analysis included all cases that had completed follow-up (that is, with a result of the confirmatory test, either positive or negative) up to January 2020, regardless of compliance. The per-protocol (PP) analysis included all cases that had finished follow-up and had taken at least 90% of the treatment drugs, as defined in the protocol. Head-to-head comparisons were made between statin and non-statin users for different levels of treatment: first for empirical vs. culture-guided cohorts globally, then for first and rescue eradication attempts (both in empirical and culture-guided cohorts), and finally for each of the most frequently used therapies (only therapies administered to at least 40 patients globally [including both statin users and non-users] were specifically evaluated).

Head-to-head comparisons of treatment effectiveness were made according to the type of statin used: simvastatin, atorvastatin, rosuvastatin, and other statins (including fluvastatin, pitavastatin, pravastatin, and lovastatin).

### 4.3. Safety Analysis

Head-to-head comparisons of AE incidence were made between statin and non-statin users as follows: first in empirical vs. culture-guided cohorts globally, then in first and rescue eradication attempts (both in empirical and culture-guided cohorts), and finally in each of the most frequently used therapies (only therapies administered to at least 40 patients globally [including both statin users and non-users] were specifically evaluated).

Head-to-head comparisons of safety were also performed overall and considering the aforementioned four statin categories.

### 4.4. Statistical Analyses

Continuous variables were presented as mean and standard deviation. Qualitative variables were reported as percentages and 95% confidence intervals (95%CI). Differences between groups were analyzed with the Chi-square test. A univariate analysis was performed evaluating mITT effectiveness or safety and the presence or absence of statins in the therapy. A multivariate analysis was also carried out using a logistic regression model by means of the stepwise forward likelihood method with *H. pylori* mITT eradication a as dependent variable, and age, gender, treatment duration, PPI doses, compliance and statin´s presence/absence, as independent factors. Significance was considered at *p* < 0.05.

## 5. Conclusions

In conclusion, our study provides a comprehensive global overview of the inconsistent usefulness of adding statins to eradication therapies against *H. pylori* as a strategy to increase effectiveness. Therefore, the addition of statins to the *H. pylori* treatment cannot be recommended to improve eradication. However, ideally, large randomized studies should be specifically performed to definitively explore this option. Nonetheless, it is important to remark that currently recommended regimens are more complex than those recommended several years ago, showing a tendency to use quadruple instead of triple therapies, advocating for extended duration, and using high PPI doses. The addition of another medication –with the aim of increasing *H. pylori* eradication– has to be carefully weighed as this could affect treatment adherence, which is a key factor affecting the effectiveness.

## Figures and Tables

**Figure 1 antibiotics-10-00965-f001:**
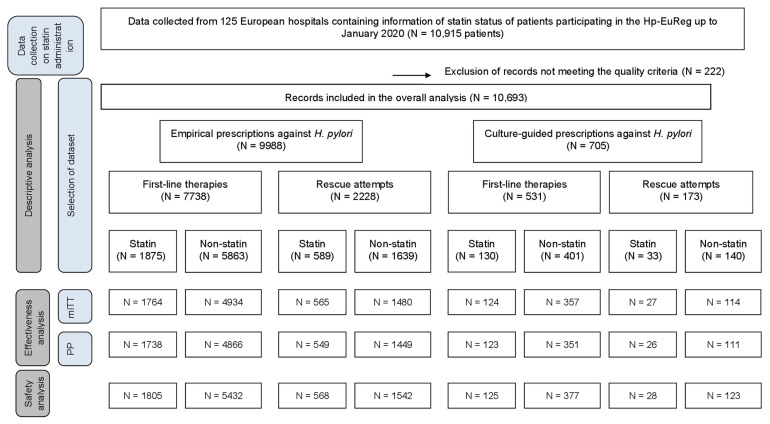
Flowchart of the patients included. N = number of patients treated. mITT = modified intention-to-treat analysis. PP = per protocol analysis.

**Figure 2 antibiotics-10-00965-f002:**
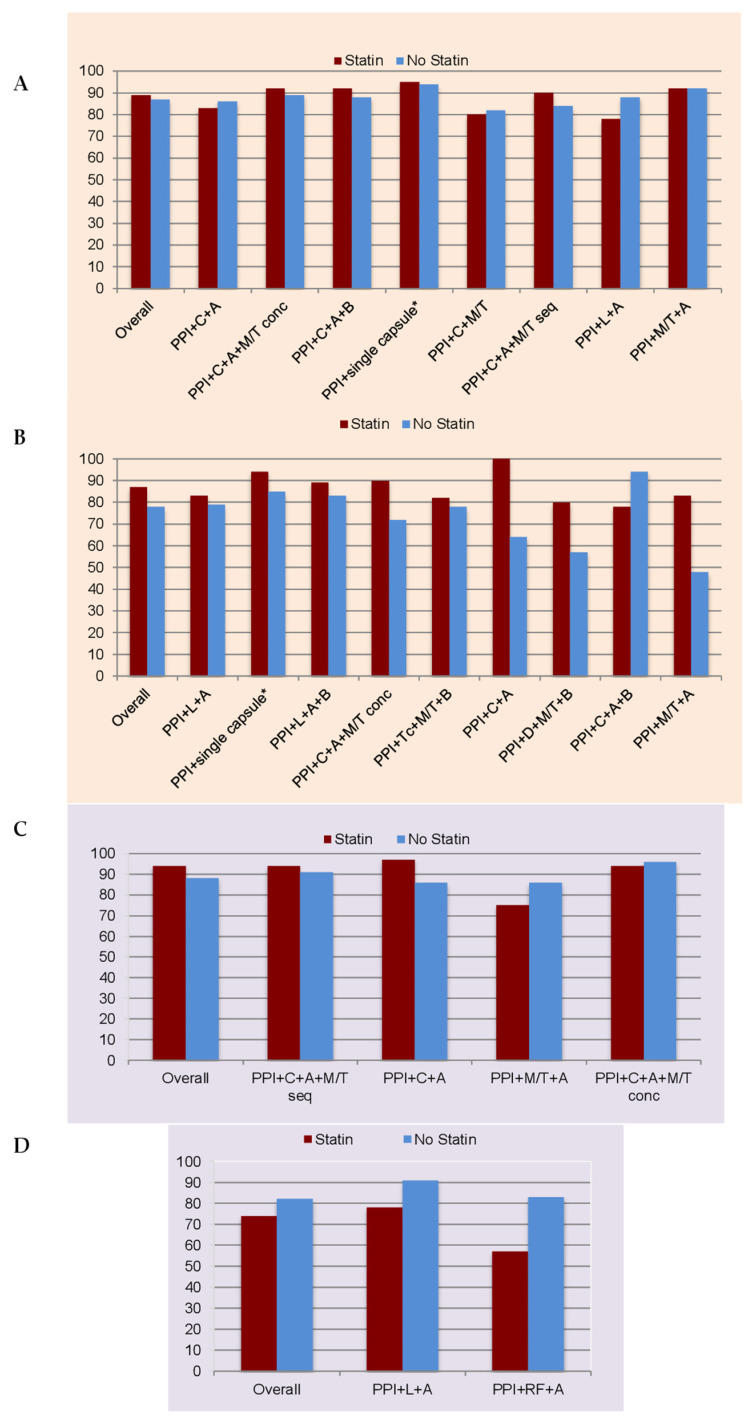
mITT effectiveness according to statin-status and receiving each specific therapy. (**A**) Empirical approach: first-line. (**B**) Empirical approach: rescue lines. (**C**) Culture-guided approach: first-line. (**D**) Culture-guided approach: rescue lines. PPI = proton pump inhibitor. C = clarithromycin. A = amoxicillin. M = metronidazole. T = tinidazole. B = bismuth. Single-capsule * = three-in-one single capsule containing bismuth, tetracycline and metronidazole. L = levofloxacin. Tc = tetracycline. D = doxycycline. RF = rifabutine. Conc = concomitant administration. Seq = sequential administration.

**Figure 3 antibiotics-10-00965-f003:**
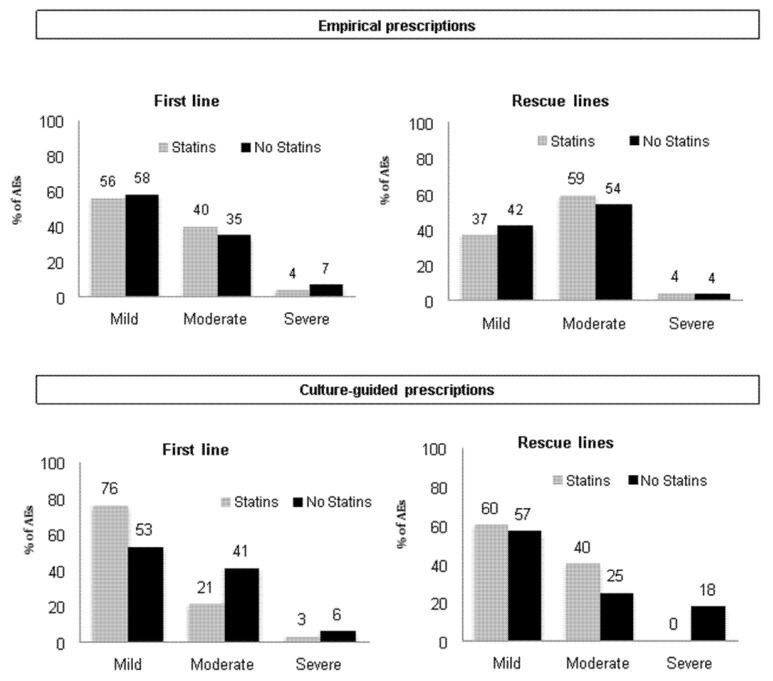
Distribution of adverse event severity presented according to statin use and treatment line.

**Table 1 antibiotics-10-00965-t001:** Demographic analysis and use of statins.

		Statin Users, N (%)	Non-Statin Users, N (%)	*p*-Value
Patients evaluated		2635 (25)	8058 (75)	
Age (years) +/−SD		63 ± 10	53 ± 15	**<0.001**
Gender	Females	1518 (58)	5093 (63)	**<0.001**
	Males	1114 (42)	2962 (37)	
Penicillin allergy	Yes	141 (5)	347 (4)	**0.026**
	No	2494 (95)	7711 (96)	
Indication	No ulcer	1982 (75.3)	6526 (81)	**<0.001**
	Ulcer disease	649 (25)	1511 (19)	
Line of eradication	First line	2005 (76)	6264 (78)	0.097
	Rescue lines	622 (24)	1779 (22)	
Prescription	Empirical	2472 (94)	7516 (93)	0.332
	Culture-guided	163 (6)	542 (7)	
PPI dose	Low	949 (37)	3220 (41)	**0.002**
	Standard	626 (24)	1742 (22)	
	High	1006 (39)	2914 (37)	
Length (days)	7	219 (9)	941 (12)	**0.002**
	10	1425 (55)	4195 (53)	
	14	929 (36)	2719 (35)	
Compliance (≥90%)	Yes	2476 (98)	7262 (98)	0.502
	No	54 (2)	176 (2)	
Adverse events	Yes	594 (24)	1703 (23)	0.460
	No	1934 (76)	5772 (77)	
Effectiveness	mITT	2190 (88)	5874 (85)	**<0.001**
	PP	2160 (89)	5818 (86)	**0.001**

N = number of patients included. % = proportion of patients included. SD = standard deviation. Low dose PPI = 20 mg omeprazole equivalents, two times per day, standard dose PPI = 40 mg omeprazole equivalents, two times per day, high dose PPI = 60 mg omeprazole equivalents, two times per day. mITT = modified-intention-to-treat effectiveness. PP = per protocol effectiveness. Significant *p*-values are highlighted in bold.

**Table 2 antibiotics-10-00965-t002:** Demographic analysis of the statin cohort stratified by type of statin.

		Simvastatin N (%)	Atorvastatin N (%)	Rosuvastatin N (%)	Other N (%)	*p*-Value
Patients evaluated		155 (45)	134 (39)	37 (11)	17 (5)	
Age (years) +/−SD		63 ± 10	63 ± 10	66 ± 8	63 ± 13	0.492
Gender	Females	86 (55.5)	69 (51.5)	19 (51)	10 (59)	0.869
	Males	69 (44.5)	65 (48.5)	18 (49)	7 (41)	
Penicillin allergy	Yes	7 (4.5)	4 (3)	0 (0)	1 (6)	0.532
	No	148 (95.5)	130 (97)	37 (100)	16 (94)	
Indication	No ulcer	119 (77)	91 (68)	32 (86.5)	14 (82)	0.070
	Ulcer disease	35 (23)	43 (32)	5 (13.5)	3 (18)	
Treatment attempt	First	120 (77)	107 (81)	30 (81)	14 (82)	0.862
	Rescue	35 (23)	25 (19)	7 (19)	3 (18)	
Prescription	Empirical	135 (87)	108 (81)	35 (95)	15 (88)	0.142
	Culture-guided	20 (13)	26 (19)	2 (5)	2 (12)	
PPI dose	Low	75 (49)	80 (62)	29 (78)	10 (59)	**0.026**
	Standard	43 (28)	23 (18)	2 (5)	4 (23)	
	High	36 (23)	26 (20)	6 (16)	3 (18)	
Length (days)	7	17 (11)	17 (13)	5 (13)	2 (12)	0.278
	10	107 (70)	90 (69)	31 (84)	11 (65)	
	14	28 (18)	23 (18)	1 (3)	4 (23)	
Compliance (≥90%)	Yes	148 (98)	128 (98.5)	37 (100)	16 (100)	0.789
	No	3 (2)	2 (1.5)	0 (0)	0 (0)	
Adverse events	Yes	18 (12)	21 (16)	5 (13.5)	2 (12.5)	0.751
	No	136 (88)	109 (84)	32 (86.5)	14 (87.5)	
Effectiveness	mITT	122 (84)	112 (91)	29 (81)	16 (100)	0.086
	PP	122 (85)	111 (92)	29 (81)	16 (100)	0.074

N = number of patients included. % = proportion of patients included. Others = includes pravastatin, pitavastatin, lovastatin and fluvastatin. SD = standard deviation. Low dose PPI = 20 mg omeprazole equivalents, two times per day, standard dose PPI = 40 mg omeprazole equivalents, two times per day, high dose PPI = 60 mg omeprazole equivalents, two times per day. mITT = modified-intention-to-treat effectiveness. PP = per protocol effectiveness. Significant *p*-values are highlighted in bold.

**Table 3 antibiotics-10-00965-t003:** Univariate analysis of effectiveness and safety according to the type of prescription (empirical versus culture-guided).

	N (%)	mITT Effectiveness			PP Effectiveness			Adverse Events		
		N Total (%)	95% CI	*p*-Value	N Total (%)	95% CI	*p*-Value	N Total (%)	95% CI	*p*-Value
Overall empirical therapies										
Statins	2472 (25)	2330 (88)	87–89	<0.001	2288 (88.5)	87–90	0.001	2375 (24)	22–25	0.542
No S	7516 (75)	6414 (85)	84–86		6315 (86)	85–87		6975 (23)	22–24	
Overall culture-guided therapies										
Statins	163 (23)	151 (90)	84–94	0.266	149 (91)	85–95	0.330	153 (22)	16–30	0.589
No S	542 (77)	471 (87)	83–90		462 (88)	84–91		500 (20)	17–24	

N = number of patients included. % = percentage of patients included. mITT = modified intention-to-treat. PP = per protocol. 95% CI = 95% confidence interval. S = statins. Significant *p*-values are highlighted in bold.

**Table 4 antibiotics-10-00965-t004:** Multivariate analysis according to mITT effectiveness in empirical versus culture-guided prescriptions.

		Empirical Prescription N = 9988			Culture-Guided Prescription N = 705		
		OR	95% CI	*p*-Value	OR	95% CI	*p*-Value
Gender	Female	1			NS		
	Male	1.218	1.1–1.4	0.004			
Indication	No ulcer	1			NS		
	Ulcer disease	1.272	1.1–1.5	0.005			
Treatment attempt	First	1			1		
	Rescue	0.522	0.45–0.60	<0.001	0.480	0.29–0.81	0.006
Length (days)	7	1			NS		
	10	1.375	1.1–1.7	0.001			
	14	1.424	1.1–1.8	0.002			
PPI dose	Low	1			NS		
	Standard	1.492	1.3–1.8	<0.001			
	High	2.026	1.7–2.4	<0.001			
Compliance	No	1			1		
	Yes	5.224	3.6–7.6	<0.001	12.716	3.6–45	<0.001
Statin use	No	1			NS		
	Yes	1.269	1.1–1.5	0.002			

mITT = modified intention-to-treat. OR = odds ratio. 95% CI = 95% confidence interval. PPI = proton pump inhibitor. Low dose PPI = 20 mg omeprazole equivalents, two times per day. Standard dose PPI = 40 mg omeprazole equivalents, two times per day. High dose PPI = 60 mg omeprazole equivalents, two times per day. NS = statistically not significant.

**Table 5 antibiotics-10-00965-t005:** Effectiveness and safety in first-line empirically prescribed therapies.

		N (R)	mITT Effectiveness			PP Effectiveness			Adverse Events		
			N (%)	95% CI	*p*-Value	N Total (%)	95% CI	*p*-Value	N Total (%)	95% CI	*p*-Value
Overall	Statins	1875	1764 (88.5)	87–90	0.254	1738 (89)	87–90	0.388	1805 (20)	18–22	0.197
	No S	5863	4934 (87)	86–88		4866 (88)	87–89		5432 (21)	20–23	
PPI + C + A	Statins	605 (32)	541 (83)	79–86	0.063	531 (83)	79–86	**0.048**	567 (15)	12–18	0.792
	No S	2259 (39)	1779 (86)	84–87		1753 (86)	85–88		1985 (15)	14–17	
PPI + C + A + M/T conc	Statins	541 (29)	532 (91.5)	89–94	0.056	526 (92)	89–94	0.099	537 (19)	16–22	**0.004**
	No S	1253 (21)	1207 (88.5)	87–90		1189 (89)	87–91		1229 (25)	23–28	
PPI + C + A + B	Statins	186 (10)	182 (92)	87–96	0.131	177 (92)	87–96	0.180	181 (30)	24–38	0.094
	No S	840 (14)	610 (88)	86–91		603 (89)	86–91		824 (24)	22–28	
PPI + single capsule *	Statins	291 (16)	279 (95)	92–98	0.540	275 (96)	93–98	0.700	283 (26)	21–32	0.067
	No S	636 (11)	601 (94)	92–96		590 (95)	93–97		612 (32)	29–36	
PPI + C+ M/T	Statins	99 (5)	93 (80)	70–87	0.660	93 (80)	70–87	0.657	94 (19)	12–29	0.852
	No S	441 (8)	384 (81.5)	77–85		382 (82)	77–85		395 (20)	16–24	
PPI + C + A + M/T seq	Statins	45 (2)	41 (90)	77–97	0.566	40 (90)	76–97	0.566	42 (7)	2–20	0.066
	No S	79 (1)	70 (84)	74–92		70 (84)	74–92		73 (22)	13–33	
PPI + L + A	Statins	24 (1)	23 (78)	56–93	0.243	23 (78)	56–93	0.254	24 (29)	13–51	0.429
	No S	79 (1)	75 (88)	78–94		74 (88)	78–94		75 (21)	13–32	
PPI + M/T + A	Statins	13 (1)	13 (92)	64–100	1	13 (92)	64–100	1	13 (31)	9–61	0.733
	No S	55 (1)	47 (91.5)	80–98		47 (91.5)	80–98		55 (25.5)	15–39	

N = number of patients included. R = proportion of patients receiving each therapy according to the statin/no statin cohort. % = proportion of patients presenting effectiveness or the adverse event. mITT = modified intention-to-treat. PP = per protocol. 95% CI = 95% confidence interval. S = statins. PPI = proton pump inhibitor. C = clarithromycin. A = amoxicillin. M = metronidazole. T = tinidazole. B = bismuth. Single capsule * = three-in-one single capsule containing bismuth, tetracycline, and metronidazole. L = levofloxacin. Conc = concomitant administration. Seq = sequential administration. Significant *p*-values are highlighted in bold.

**Table 6 antibiotics-10-00965-t006:** Multivariate analysis of mITT effectiveness in first-line empirically prescribed therapies.

		Overall			PPI + C+A			PPI + C+M/T			PPI + C+A + M/T Conc			PPI + C+A + B			PPI + Single Capsule *		
		OR	95% CI	*p*-Value	OR	95% CI	*p*-Value	OR	95% CI	*p*-Value	OR	95% CI	*p*-Value	OR	95% CI	*p*-Value	OR	95% CI	*p*-Value
Gender	Female	1						NS			NS			NS			NS		
	Male	1.367	1.2–1.6	<0.001	1.547	1.2–1.9	0.001												
Length (days)	7	1			NS						NS			NS			NS		
	10	1.328	1.1–1.6	0.008				0.343	0.15–0.78	0.011									
	14	1.286	1–1.7	0.047				0.261	0.12–0.58	0.001									
PPI dose	Low	1									1			1			NS		
	Standard	1.619	1.3–2	<0.001	1.843	1.4–2.4	<0.001	3.698	1.2–12	0.026	1.381	0.92–2.1	0.122	7.451	3.0–18	<0.001			
	High	1.982	1.6–2.4	<0.001	2.644	1.9–3.6	<0.001	2.518	1.4–4.7	0.003	1.827	1.3–2.6	0.001	2.451	1.5–4.0	<0.001			
Compliance	No	1						NS			1			NS			1		
	Yes	6.197	4–9.7	<0.001	5.613	2.6–12.1	<0.001				10.136	4.4–24	<0.001				29.0	9.6–88	<0.001
Statin use	No	NS						NS			NS			NS			NS		
	Yes				0.763	0.59–0.99	0.046												

mITT = modified intention-to-treat. PPI = proton pump inhibitor. C = clarithromycin. A = amoxicillin. M = metronidazole. T = tinidazole. B = bismuth. Single capsule * = three-in-one single capsule containing bismuth, tetracycline, and metronidazole. Conc = concomitant administration. OR = odds ratio. 95% CI = 95% confidence interval. Low dose PPI = 20 mg omeprazole equivalents, two times per day. Standard dose PPI = 40 mg omeprazole equivalents, two times per day. High dose PPI = 60 mg omeprazole equivalents, two times per day. NS = statistically not significant.

**Table 7 antibiotics-10-00965-t007:** Effectiveness and safety in rescue lines empirically prescribed therapies: 2nd–6th lines.

		N (R)	mITT Effectiveness			PP Effectiveness			Adverse Events		
			N (%)	95% CI	*p*-Value	N (%)	95% CI	*p*-Value	N (%)	95% CI	*p*-Value
Overall	Statins No S	589 1639	565 (87) 1480 (78)	84–90 76–80	**<0.001**	549 (88) 1449 (78)	85–91 76–80	**<0.001**	568 (35) 1542 (28)	31–39 26–31	**0.003**
PPI + L + A	Statins No S	177 (30) 545 (33)	172 (83) 504 (79)	76–88 75–82	0.263	169 (83) 494 (79)	76–88 75–83	0.275	175 (27) 521 (25)	21–35 22–29	0.584
PPI + single capsule *	Statins No S	170 (29) 328 (20)	164 (94) 298 (85)	89–97 81–89	**0.006**	159 (95) 291 (86)	90–98 81–89	**0.002**	165 (42) 310 (39)	35–50 34–45	0.473
PPI + L + A + B	Statins No S	65 (11) 178 (11)	63 (89) 166 (83)	78–95 77–89	0.280	59 (91.5) 164 (83.5)	81–97 77–89	0.133	62 (39) 168 (34.5)	27–52 27–42	0.556
PPI + C + A + M/T conc	Statins	31 (5)	31 (90)	74–98	**0.043**	30 (90)	74–98	0.107	31 (29)	14–48	0.253
	No S	76 (5)	71 (72)	60–82		68 (73.5)	61–84		74 (19)	11–30	
PPI + M/T + Tc + B	Statins	17 (3)	17 (82)	57–96	1	17 (82)	57–96	1	17 (35)	14–62	0.933
	No S	88 (5)	83 (78)	68–87		82 (78)	68–86		88 (36)	26–47	
PPI + C + A	Statins	10 (2)	10 (100)	NA	**0.024**	10 (100)	NA	**0.024**	10 (20)	3–56	1
	No S	53 (3)	44 (64)	48–78		44 (64)	48–78		52 (23)	13–37	
PPI + M/T + D + B	Statins	15 (3)	15 (80)	52–96	0.134	14 (79)	49–95	0.205	14 (29)	8–58	0.728
	No S	44 (3)	42 (57)	41–72		41 (56)	40–72		43 (23)	12–39	
PPI + C + A + B	Statins	10 (2)	9 (78)	40–97	0.188	9 (78)	40–97	0.111	9 (56)	21–86	**0.043**
	No S	40 (2)	34 (94)	80–99		33 (97)	84–100		36 (19)	8–36	
PPI + M/T + A	Statins	6 (1)	6 (83)	36–100	0.187	6 (83)	36–100	0.187	6 (0)	0	1
	No S	35 (2)	29 (48)	30–68		29 (48)	30–68		31 (6.5)	1–21	

mITT = modified intention-to-treat. PP = per protocol. N = number of patients included. R = proportion of patients receiving each therapy according to the statin/no statin cohort. % = proportion of patients presenting success or adverse events with the eradication therapy. 95% CI = 95% confidence interval. S = statins. PPI = proton pump inhibitor. L = levofloxacin. A = amoxicillin. Single capsule * = three-in-one single capsule containing bismuth, tetracycline and metronidazole. B = bismuth. C = clarithromycin. M = metronidazole. T = tinidazole. Tc = tetracycline. D = doxycycline. Conc = concomitant administration. Significant *p*-values are highlighted in bold.

**Table 8 antibiotics-10-00965-t008:** Multivariate analysis of mITT effectiveness in rescue attempts according to empirically prescribed therapies: 2nd–6th lines.

		Overall			PPI + L+A			PPI + Single Capsule *			PPI + L+A + B			PPI + C+A + M/T Conc			PPI + M/T + Tc + B		
		OR	95% CI	*p*-Value	OR	95% CI	*p*-Value	OR	95% CI	*p*-Value	OR	95% CI	*p*-Value	OR	95% CI	*p*-Value	OR	95% CI	*p*-Value
Age (years)	18–30	1			NS			NS			NS			NS			NS		
	31–50	0.413	0.20–0.84	0.015															
	51–70	0.357	0.18–0.72	0.004															
	>70	0.442	0.20–0.96	0.039															
Diagnosis	No ulcer	1			NS			NS			NS			NS			NS		
	Ulcer disease	1.520	1.1–2.1	0.008															
Length (days)	7	1						NS			NS			NS			1		
	10	2.425	1.4–4.2	0.001	1												1.246	0.26–6.0	0.784
	14	2.914	1.6–5.2	<0.001	2.818	1.6–5.0	<0.001										5.640	1.0–31	0.046
PPI dose	Low	1			1			NS			1			NS			NS		
	Standard	1.204	0.89–1.6	0.230	1.283	0.80–2.1	0.296				2.778	0.43–18	0.284						
	High	1.995	1.5–2.7	<0.001	1.921	1.1–3.4	0.024				3.822	1.2–12	0.025						
Compliance	No	1			NS			1			1			NS			NS		
	Yes	3.937	2.0–7.6	<0.001				4.442	1.3–15.8	0.021	6.880	1.3–36	0.022						
Statin use	No	1			NS			1			NS			1			NS		
	Yes	1.896	1.4–2.6	<0.001				2.752	1.3–5.7	0.006				3.660	0.99–13	0.050			

mITT = modified intention-to-treat. PPI = proton pump inhibitor. L = levofloxacin. A = amoxicillin. Single capsule * = three-in-one single capsule containing bismuth, tetracycline and metronidazole. B = bismuth. C = clarithromycin. M = metronidazole. T = tinidazole. Conc = concomitant administration. Tc = tetracycline. OR = odds ratio. 95% CI = 95% confidence interval. Low dose PPI = 20 mg omeprazole equivalents, two times per day. Standard dose PPI = 40 mg omeprazole equivalents, two times per day. High dose PPI = 60 mg omeprazole equivalents, two times per day. NS = statistically not significant.

**Table 9 antibiotics-10-00965-t009:** Effectiveness and safety in culture-guided eradication therapies.

		N (R)	mITT Effectiveness			PP Effectiveness			Adverse Events		
			N (%)	95% CI	*p*-Value	N (%)	95% CI	*p*-Value	N (%)	95% CI	*p*-Value
First-line (N = 531)											
Overall	Statins	130	124 (93.5)	88–97	0.095	123 (93.5)	88–97	0.163	125 (21)	14–29	0.871
	No S	401	357 (88)	84–91		351 (89)	85–92		377 (22)	18–26	
PPI + C + A + M/T seq	Statins	54 (42)	53 (94)	84–99	0.557	52 (94)	84–99	0.559	53 (23)	12–36	0.794
	No S	143 (36)	123 (91)	85.96		122 (90)	84–95		135 (24)	18–33	
PPI + C + A	Statins	40 (31)	36 (97)	86–100	0.074	36 (97)	86–100	0.189	37 (8)	2–22	0.903
	No S	131 (33)	117 (85.5)	78–91		113 (88.5)	81–94		120 (8)	4–14	
PPI + M/T + A	Statins	4 (3)	4 (75)	19–99	0.496	4 (75)	19–99	0.496	4 (50)	7–93	0.229
	No S	44 (11)	42 (86)	72–95		42 (86)	72–95		43 (21)	10–36	
PPI + C + A + M/T conc	Statins	16 (12)	16 (94)	70–100	1	16 (94)	70–100	1	16 (38)	15–65	0.950
	No S	26 (7)	25 (96)	80–100		24 (96)	79–100		26 (39)	20–59	
Rescue lines (N = 173)											
Overall	Statins	33	27 (74)	54–89	0.379	26 (77)	56–91	0.479	28 (29)	13–49	0.130
	No S	140	114 (82)	73–88		111 (83)	75–89		123 (16)	10–24	
PPI + L + A	Statins	10 (30)	9 (78)	40–97	0.299	9 (78)	40–97	0.299	8 (25)	3–65	0.578
	No S	37 (26)	32 (91)	75–98		32 (91)	75–98		33 (12)	3–28	
PPI + RF + A	Statins	7 (22)	7 (57)	18–90	0.130	7 (57)	18–90	0.143	7 (57)	18–90	**0.018**
	No S	36 (26)	30 (83)	65–94		29 (83)	64–94		34 (12)	3–28	

mITT = modified intention-to-treat. PP = per protocol. N = number of patients included. R = proportion of patients receiving each therapy according to the statin/no statin cohort. % = proportion of patients presenting success or adverse events with the eradication therapy. 95% CI = 95% confidence interval. S = statins. PPI = proton pump inhibitor. C = clarithromycin. A = amoxicillin. M = metronidazole. T = tinidazole. Conc = concomitant administration. Seq = sequential administration. L = levofloxacin. RF = rifabutine. Significant *p*-values are highlighted in bold.

**Table 10 antibiotics-10-00965-t010:** Multivariate analysis of mITT effectiveness according to culture-guided prescribed therapies.

	First-Line			Rescue Therapies		
	OR	95% CI	*p*-Value	OR	95% CI	*p*-Value
Compliance						
No	1			NS		
Yes	12.4	2.7–57	0.001			

mITT = modified intention-to-treat. Rescue therapies = embracing therapies prescribed from 2nd–6th eradication attempt. OR = odds ratio. 95% CI = 95% confidence interval. NS = non statistically significant.

## Data Availability

The data presented in this study are available on request from the corresponding author.

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
