# Peer review of "The Role of Statins on *Helicobacter pylori* Eradication: Results from the European Registry on the Management of *H. pylori* (Hp-EuReg)"

_antibiotics, 2021, doi:10.3390/antibiotics10080965_

Round 1

Reviewer 1 Report

Manuscript ID: antibiotics-1297329

This paper analyses a possible impact of statins on a range of therapeutic strategies to eradicate H.pylori; the datasets discussed here have been obtained within the Hp-EuReg observational study collecting information from >100 European hospitals from 25 different countries.

Overall, the paper is well-written and scientifically sound, and the conclusions adequately supported by the data. The cited literature is appropriate and generally up to date. I believe it could be of interest for the Journal’s readers. I have only some suggestions for the authors, as follows.

  • The type of statin is available only for 13% of patients undergoing this treatment; no information is available on dosage. Authors already identify such biases in the discussion, but I wonder whether the analysis according to the type of statin performed here is appropriate (see lines 289-293).
  • Eradication was confirmed through different methods, which may have different sensitivities. Is this a possible confounding factor? If so, please comment on this.
  • Figure 2. The information contained in these graphs can be included easily in Table 5. If feasible, it would be better to have a graphical representation of treatment effectiveness rather than on the proportions of patients receiving a certain therapy.
  • The following paper might be added to the discussion or where more appropriate:  PLoS One. 2016 Jan 5;11(1):e0146432. doi: 10.1371/journal.pone.0146432. eCollection 2016. PMID: 26730715

Reviewer 2 Report

The research article entitles as ‘’The role of statins on Helicobacter pylori eradication: results from the European Registry on the Management of H. pylori (Hp-EuReg)’’ was reviewed.

  1. pylori infection prevalence and complexities are varied in different geographical regions due to general hygiene practices, antibiotics use and resistances pattern, genetics of both bacteria and host, above all the economic status of the populations. Therefore, more than 50% of the global population still harbours H. pylori infection. The resistance to standard antibiotics therapy is a concern in almost every part of the world. Hence, application of supplementary approaches along with antibiotics, which ultimately improves the treatment regimen has relevance in the treatment of H. pylori infection.

The authors tried to bring the above-mentioned strategy in the present study. They have quoted some of the repurposing studies of statin as adjuvant for antibiotic therapy in H. pylori infection, but the results were not much satisfactory. There are reports on cholesterol depletion from host and glycosylation in H. pylori infection and link to immune evasion, dysregulated cytokine signalling and CagA translocation by T4SS. How the statin use in patients will help as an adjuvant in antimicrobial treatment with respects to the above facts are not clear. The present study employed a quite good number of populations altogether, however, subgroup analysis needs more thorough planning and needs larger groups for derivation of any proposed connection.

Comments

Minor

The study design should be improved, and subgroup representation must be increased.

The methods described in the article must be clear and understandable to the readers.

Major

The major drawback which I found is the lack of inclusion of statin concentration in the present study. The groups of patients with different concentrations of statins must be included. This will give a clear picture of concentration dependent effect of statin as an adjuvant in antimicrobial therapy.

Apart from extensive subgroup divisions with low representation, the study might concentrate on specificity of hypothesis of repurposing statin as an adjuvant of antimicrobial therapy.

I would not suggest to accept this manuscript for publication in this form.

Reviewer 3 Report

  • “Helicobacter pylori (H. pylori) is a Gram-negative bacterium involved 98 in the etiopathogenesis of several common diseases such as peptic ulcer disease, chronic gastritis or gastric cancer [1, 2].”

Cite extra-gastric manifestations too (see, for example, “A 2016 panorama of Helicobacter pylori infection: key messages for clinicians. Panminerva Med. 2016 Dec;58(4):304-317. PMID: 27716738.”)

  • “Several strategies have been proposed to optimize the success rate of eradication therapies, most of them focused on extending treatment duration, using more potent drugs to decrease gastric acidity –such as high doses of proton pump inhibitors (PPI)– or using quadruple instead of triple therapies”

In this regard, the case of double therapy with amoxicillin plus ppi is interesting: your comment would be interesting

  • “This pleiotropic effect has been associated with anti-inflammatory 124 properties involving infectious diseases or even neoplastic conditions”

Dys-immune diseases too (see, for example, “Targeting IL-10, ZO-1 gene expression and IL-6/STAT-3 trans-signalling by a combination of atorvastatin and mesalazine to enhance anti-inflammatory effects and attenuate progression of oxazolone-induced colitis. Fundam Clin Pharmacol. 2021 Feb;35(1):141-142. doi: 10.1111/fcp.12638. Epub 2020 Dec 21. PMID: 33289917.”)

  • In Table 1, Table 3, Table 4, Table 6, Table 7, Table 8 do not report p value of 0, but report the last digit or < …

  • “OR = 1.27; 95% CI: 1.1–1.5; p < 0.05”: please specify if this comes from multivariate analysis

  • Use Oxford comma

  • “Clostridium difficile infection"

Use the new term (Clostridioides)

  • “The present study was designed to evaluate whether the use of statins could modify the effectiveness or the safety of the therapies prescribed against H. pylori.”

I don’t think that the study was designed to evaluate whether the use of statins: in fact, you do not have data about the dose and you have data about the specific statin only in a minority of the patients; your study is a sub analysis of your European register
